# Investigating the Impact of Polymers on Clay Flocculation and Residual Oil Behaviour Using a 2.5D Model

**DOI:** 10.3390/polym16243494

**Published:** 2024-12-14

**Authors:** Xianda Sun, Yuchen Wang, Qiansong Guo, Zhaozhuo Ouyang, Chengwu Xu, Yangdong Cao, Tao Liu, Wenjun Ma

**Affiliations:** 1State Key Laboratory of Continental Shale Oil, Northeast Petroleum University, Daqing 163318, China; sunxianda@nepu.edu.cn (X.S.); xuchw@nepu.edu.cn (C.X.); 2Key Laboratory for Enhanced Oil & Gas Recovery of the Ministry of Education, Northeast Petroleum University, Daqing 163318, China; 210201240101@stu.nepu.edu.cn (Y.W.); guoqiansong1016@163.com (Q.G.); cyangdong0221@163.com (Y.C.); 19214590395@163.com (T.L.); 18446296533@163.com (W.M.); 3Shenyang Center of China Geological Survey, Shenyang 110034, China

**Keywords:** polymer, kaolinite, locculation, micro-visualization displacement model, fluorescence microscope, binary composite flooding

## Abstract

In the process of oilfield development, the surfactant–polymer (SP) composite system has shown significant effects in enhancing oil recovery (EOR) due to its excellent interfacial activity and viscoelastic properties. However, with the continuous increase in the volume of composite flooding injection, a decline in injection–production capacity (I/P capacity) has been observed. Through the observation of frozen core slices, it was found that during the secondary composite flooding (SCF) process, a large amount of residual oil in the form of intergranular adsorption remained in the core pores. This phenomenon suggests that the displacement efficiency of the composite flooding may be affected. Research has shown that polymers undergo flocculation reactions with clay minerals (such as kaolinite, Kln) in the reservoir, leading to the formation of high-viscosity mixtures of migrating particles and crude oil (CO). These high-viscosity mixtures accumulate in local pores, making it difficult to further displace them, which causes oil trapping and negatively affects the overall displacement efficiency of secondary composite flooding (SCF). To explore this mechanism, this study used a microscopic visualization displacement model (MVDM) and microscopy techniques to observe the migration of particles during secondary composite flooding. By using kaolinite water suspension (Kln-WS) to simulate migrating particles in the reservoir, the displacement effects of the composite flooding system on the kaolinite water suspension, crude oil, and their mixtures were observed. Experimental results showed that the polymer, acting as a flocculant, promoted the flocculation of kaolinite during the displacement process, thereby increasing the viscosity of crude oil and affecting the displacement efficiency of secondary composite flooding.

## 1. Introduction

With the exploration and development of petroleum and natural gas, oil and gas reservoirs have entered the later stage of high water-cut production [1], leading to a reduction in oil and gas resources. Domestic oil and gas production is facing tremendous pressure, and how to improve oil recovery efficiency through enhanced oil recovery (EOR) is crucial. EOR techniques mainly include chemical flooding, gas flooding, thermal recovery, and microbial enhanced oil recovery (MEOR) [2,3,4,5,6]. Most oil fields in China are suitable for chemical flooding, which is one of the most important and widely used techniques in EOR. Most chemical flooding methods are based on adding water-soluble polymers to the injection water, with partially hydrolyzed polyacrylamide (HPAM) being the most commonly used [7]. After decades of technological updates and iterations, key chemical flooding technologies have been developed, such as surfactant–polymer (SP) binary composite flooding, alkali–surfactant–polymer (ASP) ternary composite flooding, and miscible flooding technologies [8]. With the widespread use of polymer-based composite flooding, various problems have arisen. The adsorption and retention of polymers in the pore throats can lead to a decline in reservoir permeability and a change in wettability, causing reservoir damage [9]. Furthermore, as the volume of composite flooding injection increases, a decline in injection–production capacity (I/P capacity) has been observed.

Studies have found that the flocculation reaction of polymers with kaolinite plays an important role in wastewater treatment. Zou Yan [10] et al. used kaolin suspension as the subject of investigation to explore the flocculation ability of polymers. Yu Xianwei [11] used montmorillonite as a proxy for wastewater to study the flocculation performance of polymers. Zhao Xianfeng [12] used kaolinite to simulate wastewater and investigate the flocculation ability of polymers. These studies confirmed the flocculation effect of polymers as flocculants on kaolinite. Using kaolinite to simulate migrating particles in the reservoir, a high-viscosity precipitate is generated, which could be a significant reason for the decline in injection–production capacity (I/P capacity) in the later stages of polymer flooding. To validate the impact of flocculation reactions on the oil displacement process, microscopic studies were conducted, and samples were taken and observed before and after displacement using microscopy.

To better observe the flocculation reaction between the polymers and migrating particles, this study employs a microscopic visualization oil displacement model, specifically a 2.5D model. This model allows in situ observation and effective testing of the thermodynamic and flow behaviors of fluids at the micrometer and nanometer scales [13,14]. Compared to traditional physical simulation experiments, the 2.5D model offers advantages such as real-time visualization, a small sample size, lower scale limitations, and shorter measurement times. It enables the etching of true reservoir pore structures and the use of kaolinite to fill the pores, simulating real reservoir conditions to observe the reactions between the polymer and kaolinite during the polymer flooding process. However, due to limitations in the materials and processes used to create the model, the channels are not at a micrometer or nanometer scale, and the flow rate and pressure during the displacement process are not easily controlled. Therefore, other methods must be adopted to reduce errors, such as using injection pumps to mitigate the impact of injection speed on the results. In recent years, many researchers have made significant progress in the theory and technology of microscopic displacement experiments [15,16,17,18]. Through the use of the microscopic visualization model for displacement experiments, the cost of experiments has been greatly reduced, allowing for direct observation of the displacement effects under a microscope, as well as the flocculation reactions between polymers and migrating particles.

## 2. The Phenomenon of Particle Migration After Binary Composite Flooding

### 2.1. The Advantages of Binary Composite Flooding for Oil Recovery

Compared to polymer flooding (PF), surfactant–polymer (SP) binary composite flooding improves oil displacement efficiency by adding surfactants to reduce interfacial tension, making the solution more easily wettable and mitigating the emulsification effects caused by polymers during experiments [19]. This transition converts water-in-oil (W/O) emulsions into oil-in-water (O/W) emulsions, improving solubility [20,21,22]. In comparison with alkali–surfactant–polymer (ASP) ternary composite flooding, the absence of alkali in the SP formula maximizes the viscoelastic properties of polymers while reducing corrosion and scaling caused by alkali. This approach maintains ultra-low interfacial tension and achieves oil displacement efficiency comparable to ASP flooding. Moreover, it significantly reduces production costs, enhances the environmental performance of chemical flooding agents, and preserves the acidic environment of kaolinite water suspension [23]. It also avoids the scaling issues associated with alkali flooding, which could otherwise interfere with experimental observations. Therefore, SP binary composite flooding is used for subsequent experiments.
Al_4_·[Si_4_O_10_](OH)_8_·nH_2_O + OH^−^→Al(OH)_3_↓ + SiO_4_^4−^ + H_2_O

### 2.2. Particle Migration Phenomenon

After conducting the composite flooding oil displacement experiment, frozen core slices were observed. It was found that clay minerals and rock fragments mixed with crude oil formed intergranular adsorbed residual oil, which accounted for more than 70% of the total residual oil.

Figure 1 illustrates the intergranular adsorbed residual oil. This phenomenon typically occurs in samples with high mud content or medium-to-high permeability, where clay minerals and rock fragments migrate and mix with crude oil, accumulating locally. This leads to changes in viscosity, reduced fluidity, and the formation of intergranular adsorbed residual oil [24]. This is primarily due to the polymer acting as a flocculant, promoting the flocculation of migrating particles such as clay, which results in precipitation and the formation of the aforementioned phenomenon. The presence of intergranular adsorbed residual oil can be observed in both three-dimensional CT images (see Figure 2a) and fluorescence photographs (see Figure 2b).

### 2.3. The Mechanism of the Particle Migration Phenomenon

In this study, kaolinite water suspension was used to simulate migrating particles, such as clay, in the reservoir. Kaolinite is the main clay mineral found in acidic soils and is one of the most typical and abundant clay minerals in ion-type rare earth ores. Its theoretical chemical composition is Al_4_Si_4_O_10_(OH)_8_ (A_12_O_3_ 41.2%; Si_2_O_4_ 8.0%; H_2_O 10.8%), and the mineral surface carries a negative charge [25,26,27]. Flocculants primarily refer to groups with positive (or negative) charges that interact with positively (or negatively) charged suspended particles in water. Through synergistic actions such as electrostatic neutralization, adsorption bridging, compression of the electric double layer, and network capture, the suspended particles adsorbed on the surface of the flocculant molecules or encapsulated by flocculants with a “core–shell” structure are agglomerated, leading to flocculation and precipitation [28]. As a flocculant, polymers cause fine particles and suspended particles in the formation to collide, forming flocculates. These flocculates remain in the formation of pores and throats as precipitates, further damaging the reservoir. Figure 3 illustrates flocculation-induced precipitation, where the viscosity significantly increases after the formation of the flocculates.

## 3. Micro-Visualization Experiment

### 3.1. Experimental Chemicals

Kaolinite water suspension (concentration: 30%); water (mineralization degree: 486); HPAM (1000 mg/L, molecular weight: 25 million); surfactant (0.20% petroleum sulfonate); crude oil (density: 0.8626 g/cm^3^, kinematic viscosity: 18.65 mm^2^/s); alcohol, acetone, deionized developing solution; hydrofluoric acid and UV glue(All experimental chemicals were sourced from Daqing City, Heilongjiang Province, China.).

### 3.2. Experimental Apparatus

Laser confocal microscope; micro-injection pump, injection syringe; fluorescence microscope; valve; micro-visualization model holder; micro-visualization model; heating plate; parallel light source; liquid recovery collection container; and image acquisition computer.

As shown in Figure 4, the prepared etched glass model was placed in a holder, with a heating plate underneath to provide a constant temperature environment. A micro-syringe pump was used to inject the displacement fluid at a uniform rate. During the displacement process, a microscope was used for observation, and images were captured via a computer for better monitoring of the displacement state. After the displacement was completed, the microscope was used again to observe the results, allowing clear visualization of the displacement efficiency and phenomena occurring during the process (All equipment and instruments were manufactured in Heilongjiang Province, China).

### 3.3. Preparation of Micro-Visualization Model

First, select the glass slide to be etched, wash it with alcohol and acetone, then clean it with deionized water and dry it in an oven at 50 °C. Second, use a laser confocal scanner to scan a 1 cm × 2 cm area of the cast film (as shown in Figure 5a). Stitch the images together and use vectorization software to convert the stitched bitmap into a vector image (as shown in Figure 5b). Add the injection and extraction holes and set aside. Third, apply a photosensitive resist to the slide by placing a drop of resin on the dried slide and using a spin coater at 4500 rpm. Set the temperature to 40 °C for 10 min. If a spin coater is unavailable, use a glass rod to spread the resin evenly, and then use another slide to scrape off the excess, ensuring a thin, uniform layer. Leave the slide horizontally in an oven at 50 °C for 1 h, then remove it and place it on a hot plate at 90 °C for 3 min, followed by 150 °C for 8 min. Set aside in a dark place to cool. Fourth, attach the prepared mask to the surface of the photosensitive film and expose it to ultraviolet light for 1 h. After exposure, quickly cure it on a hot plate at 90 °C for 6 min. Fifth, develop the slide by immersing it in the developer solution for 10 s, then rinse it with water. Inspect under a microscope to check if the development is successful. If there are areas with incomplete development, re-develop until the glass surface is fully exposed. Sixth, etch the slide using wet chemical etching, as it is faster and suitable for uniform etching over a larger area. However, as hydrofluoric acid is highly corrosive, ensure proper safety measures are in place. Place the developed slide in hydrofluoric acid for 30 min, then remove and rinse thoroughly with water. Seventh, prepare a cover glass by selecting a suitable piece, cleaning it with deionized water, and drying it at 50 °C in the oven. Apply a uniform layer of UV adhesive on the surface and press the etched glass onto it. Expose to ultraviolet light for 1 h to bond the two pieces. Eighth, drill holes using a 1 mm diameter glass drill bit at the injection and extraction ends.

Model Production Process: The adhesive layer should not be too thick, and the development process must ensure that the glass bottom surface remains visible. Care should be taken with the UV exposure time to avoid affecting the surface hardness of the adhesive. After exposure, immediate curing on the hotplate is necessary to prevent fading. A metal plate should be added on top of the hotplate to ensure even heat distribution. Excessive heat can damage the adhesive’s properties. Oversoaking the model in the developer solution for too long will cause the adhesive to peel off, so the soaking time should be controlled.

The final microscopic visualization displacement model is shown in Figure 6. The left side represents the inlet where the displacement fluid enters, while the right side represents the outlet where the fluid is discharged.

### 3.4. Preparation of Experimental Reagents

Preparation of Binary Composite Displacement Fluid: The displacement fluid was composed of a polymer solution and a surfactant solution. The polymer selected was polyacrylamide, and the surfactant chosen was petroleum sulfonate. A certain amount of polyacrylamide with a molecular weight of 25 million was dissolved in deionized water, using a stirrer to ensure thorough mixing. The temperature was maintained between 20 and 30 °C, and the pH was kept in the neutral range to improve the dissolution efficiency and stability of the polymer. The final polymer solution had a concentration of 1500 mg/L. The required surfactant was added to deionized water and stirred until fully dissolved, adjusting the surfactant concentration to approximately 0.2%. The polymer solution was then mixed with the surfactant solution, ensuring a slow addition to avoid adverse reactions or precipitation with the polymer, and the mixture was stirred thoroughly. The final composite displacement fluid was stored in a sealed container to prevent the influence of environmental factors such as high temperatures and direct sunlight.

Preparation of Kaolinite Water Suspension: High-purity kaolin was selected, dried, and ground to ensure fine particles for easy dispersion. A certain amount of deionized water was added, and the kaolin concentration was adjusted to 30%. An ultrasonic processor was used for stirring to ensure full dispersion of the kaolin, preventing aggregation. The pH of the suspension was adjusted to 4, and large particle impurities were removed by filtration to obtain a more uniform kaolin suspension. Finally, a stability test was conducted to observe the sedimentation behavior of the suspension over time. A certain amount of surfactant was added to improve the stability of the suspension.

### 3.5. Microscopic Visualization Displacement Experiment

First, the glass-etched thin slices, saturated with kaolinite water suspension, were sampled using transmitted light microscopy. The image showing the effect after saturation with the kaolinite solution is shown in the figure. It can be observed that the kaolinite water suspension filled the throats and pores of the glass-etched model (see Figure 7).

After saturating the model with kaolinite aqueous suspension, displacement experiments were conducted, including waterflooding as the control group and binary composite flooding as the experimental group. The phenomena after displacement were analyzed.

Experiment 1 Displacement Process: The micro-glass etching model was fixed in a micro-visualization holder and subjected to a vacuum treatment. Kaolinite aqueous suspension (30% concentration) was injected from the inlet, and three displacement experiments were conducted. In Experiment 1, a heating plate was used to maintain a constant temperature of 46 °C. Water (with a mineralization degree of 486) and binary composite flooding (comprising 1000 mg/L HPAM with a molecular weight of 25 million and 0.20% petroleum sulfonate as a surfactant) were injected using a syringe. The displacement was conducted using the exhaust method, with volumes of 10 PV and 20 PV, respectively. During the displacement, the flocculation effect of the polymer on the migrating particles was observed.

After Experiment 1, to demonstrate the effect of the polymer during the oil displacement process, further experiments were carried out to observe the impact of binary composite flooding on oil displacement in real reservoir conditions. The model was saturated with crude oil before continuing. Experiment 2 Displacement Process: The kaolinite aqueous suspension (30% concentration) was first injected into the model from the inlet to saturate the glass etching pores, followed by the injection of crude oil for saturation. This setup was used to simulate the effect of binary composite flooding during crude oil saturation. Binary composite flooding solutions (polymer and surfactant mixture) with volumes of 10 PV, 20 PV, and 80 PV were injected, and the displacement process was observed.

Finally, Experiment 3 was conducted, where the glass etching model was first saturated with a mixture of kaolinite (30% concentration) and crude oil, simulating a real reservoir environment. Binary composite flooding was then applied for displacement, and the results were compared with Experiment 2 to verify the impact of kaolinite’s flocculation reaction on the oil displacement efficiency of binary composite flooding. The exhaust method was again used, with volumes of 10 PV, 20 PV, and 80 PV injected for each test. This series of experiments aimed to analyze the effects of polymer flocculation and binary composite flooding on oil recovery, especially under conditions mimicking real reservoir environments.

## 4. Results Analysis

### 4.1. Experimental Results Analysis

In Experiment 1, the oil displacement effects of waterflooding and binary composite flooding were compared at injection volumes of 10 PV and 20 PV, respectively, where the black areas represent kaolinite.

As shown in Figure 8, when the injection volume is 10 PV, water flooding exhibits a low displacement efficiency, with a large amount of kaolinite suspension not effectively displaced. This indicates that water flooding has limitations in cleaning residual materials. In contrast, binary composite flooding shows significantly better performance than water flooding, effectively displacing a large amount of kaolinite suspension and leaving minimal residual material, demonstrating its significant advantage in improving recovery efficiency. The binary composite flooding not only achieves a higher displacement efficiency within a shorter injection volume but also has a clear advantage in better-removing fluids from the pores and reducing residual oil retention, thus enhancing the recovery rate.

When the injection volume increased to 20 PV, water flooding could nearly clean all the kaolinite from the pores, with only a small amount remaining in the pore corners, throats, and pore edges. This suggests that as the injection volume increased, the displacement effect of water flooding improved significantly, approaching a saturated state, and the quantity of residual material greatly decreased. Meanwhile, for the binary composite flooding, at 20 PV, the displacement effect did not show significant improvement compared to the 10 PV case, indicating that by 10 PV, the displacement process was largely complete, and further injection did not significantly drive the remaining oil to move. Additionally, although binary composite flooding, like water flooding, did not completely remove all the residual oil, its displacement effect was still significantly better than that of water flooding. This shows that, at high injection volumes, the further displacement effect of binary composite flooding gradually leveled off. Overall, the binary composite flooding demonstrates strong displacement capability at relatively low injection volumes, significantly improving the recovery rate, whereas water flooding, at higher injection volumes, effectively displaces material in the pores, but still leaves some residual oil.

Based on the results of Experiment 1, the following conclusions can be drawn. Using water flooding as a comparative experiment, binary composite flooding shows a more significant effect in the early stages in improving recovery. However, as the injection volume continues to increase, the displacement effect gradually weakens and does not significantly improve the recovery rate, even negatively affecting the injection–production capacity. This phenomenon can be attributed to the flocculation effect of the polymer, which causes kaolinite particles to precipitate and accumulate in the throats, pore corners, and edges, forming deposits that block the flow channels. After the channels are blocked, the flow of displacement fluids is restricted, leading to impaired pore connectivity and making it difficult to further displace the residual oil. In addition, the blocking of pores not only restricts fluid permeability but may also lead to a decrease in reservoir permeability and changes in wettability, causing reservoir damage. The accumulation of polymer precipitates prevents the fluid from passing through the pores smoothly, thereby negatively impacting the displacement efficiency of crude oil.

To further verify the impact of migrating particle precipitation on the displacement process, future research should conduct saturated oil experiments to observe the specific effects of different particle precipitations on displacement performance and assess the mechanisms of polymer and kaolinite particle precipitation. This will provide theoretical support for optimizing displacement strategies.

Based on the results shown in Figure 9, the following conclusions can be drawn. When the injection volume is 10 PV, the displacement effect of the binary composite flooding is not obvious, and there is still a significant amount of residual oil. This indicates that at this relatively low injection volume, the displacement fluid has not sufficiently mobilized the crude oil flow, and it has not effectively removed the residual oil from the pores, resulting in a relatively low displacement efficiency. However, as the injection volume increased to 20 PV, the displacement effect improved significantly. The injected fluid can drive a large amount of crude oil flow, accelerating the displacement process, and the amount of residual oil decreases substantially. This indicates that at this injection volume, binary composite flooding demonstrates strong displacement capability and can significantly improve the recovery rate.

When the injection volume was increased further to 80 PV, the displacement effect did not improve significantly compared to the 20 PV case. The crude oil was mostly displaced, with the remaining oil primarily concentrated in hard-to-reach areas such as pore corners, throats, and pore edges. Even at high injection volumes, a small portion of crude oil cannot be fully displaced. This suggests that the characteristics of the reservoir in these special locations, such as the complexity of the pore structure and retention phenomena, limit the effective flow of the displacement fluid, causing crude oil retention.

This experimental result shows that binary composite flooding has a good displacement effect during the oil displacement process, especially in the early stages, where it can significantly improve the recovery rate. Increasing the injection volume effectively enhances the displacement effect, but after a certain injection volume was reached, the displacement effect stabilized, and further injection did not significantly improve the displacement of residual oil. This was mainly due to the difficulty of removing remaining oil from deeper pore spaces. This phenomenon provides an important reference for optimizing subsequent oil displacement technologies, suggesting that more precise optimization measures targeting the pore structure and the specific locations of retained oil are needed to further improve recovery and displacement efficiency.

Based on the experimental results shown in Figure 10, the following conclusions can be drawn. When the injection volume was 10 PV, a large amount of mixed liquid remained in the experiment, indicating that at this relatively low injection volume, the binary composite flooding did not show significant oil displacement effects. It failed to effectively remove crude oil and mixed liquid from the pores, and a considerable amount of residual liquid remains. This suggests that the injection volume was insufficient to fully drive fluid flow, preventing effective oil and gas displacement, and the displacement process remained somewhat limited.

When the injection volume increased to 20 PV, the binary composite flooding could displace part of the mixed liquid, resulting in some improvement in the displacement effect. The distribution of crude oil and kaolinite became more uniform, and the amount of residual mixed liquid decreased. However, when compared to the experimental results at the same injection volume in Experiment 2, the amount of mixed liquid in the model remained relatively large, indicating that the binary composite flooding had not yet achieved the optimal displacement effect at this injection volume. The removal of mixed liquid was still incomplete, and some regions of the reservoir had not been fully displaced.

Further increasing the injection volume to 80 PV revealed dominant flow channels and the displacement effect improved significantly; especially in the regions of high-quality channels, the displacement fluid can more effectively remove crude oil and mixed liquid. However, despite the improvement in displacement effect, when compared to Experiment 2, a large amount of mixed liquid residue could still be observed after the displacement was completed. This suggests that in certain pore regions, fluid flow remained somewhat restricted, and not all crude oil and mixed liquid had been displaced. This indicates that the flocculation reaction of kaolinite had impacted the oil displacement effect. The flocculation and precipitation caused by kaolinite had prevented the displacement fluid from fully penetrating and removing all the residual materials.

To further investigate the impact of kaolinite’s flocculation reaction on the oil displacement process, a detailed analysis of the composition of the residual mixed liquid is necessary. By using imaging techniques such as transmitted light and orthogonal light microscopy, it is possible to capture images of the same location to observe the distribution of crude oil, polymer, and kaolinite within the mixed liquid. This will help further elucidate the composition of the mixed liquid, assess the specific impact of kaolinite’s flocculation reaction on the reservoir displacement effect, and provide a theoretical basis for optimizing binary composite flooding technology.

As shown in Figure 11, through comparison of transmitted light and orthogonal light observations, it is evident that the flocculation reaction of kaolinite significantly impacted the oil displacement efficiency during binary composite flooding. The flocculation and precipitation of kaolinite in the displacement fluid led to pore plugging, forming discontinuous flow paths. These precipitates not only occupied the pores that should have been conducting fluid but also mixed with crude oil to form a high-viscosity mixture, greatly reducing the flowability of the crude oil. As a result, the crude oil became difficult to displace effectively, and it could not be efficiently pushed out by the injected fluid.

The high-viscosity mixed oil phase became trapped in narrow regions of the pore structure, such as the throats, corners, and pore edges, forming residual oil that was difficult to remove during the displacement process. This phenomenon significantly hampered the overall displacement efficiency of binary composite flooding. Despite increasing the injection volume, the high-viscosity mixture obstructed further penetration and flow of the displacement fluid, preventing the complete removal of crude oil, particularly in some inaccessible areas of the reservoir.

The precipitation of kaolinite not only restricts fluid flow but also may cause damage to the pore structure, leading to a decrease in reservoir permeability. This limits the full development potential of the oil reservoir. Furthermore, the blocking effect of the precipitates could negatively impact the long-term stability of the reservoir, potentially causing irreversible damage. Therefore, to optimize the oil displacement effect of binary composite flooding, it is crucial to thoroughly investigate the specific impact of kaolinite’s flocculation reaction on the displacement process. Effective anti-blocking technologies and methods to clear precipitates need to be explored to reduce damage to the reservoir and further enhance displacement efficiency and recovery rates.

### 4.2. Impact of Flocculation Reactions on the Oil Displacement Process and Solutions

Through experimental research, it has been observed that the flocculation reaction between polymers and kaolinite not only plays a role in wastewater treatment but also has a significant impact on polymer-dominated compound oil displacement processes. During oil displacement, the polymer interacts with kaolinite, forming high-viscosity mixtures. This high-viscosity state significantly reduces the mobility of the displacing fluid, preventing it from effectively penetrating the pores and thus affecting the displacement efficiency.

When polymer–kaolinite flocs accumulate, they tend to deposit in critical areas of the reservoir, such as pore throats, corners, and pore edges, causing pore blockage. This blockage restricts the free flow of fluid and prevents a portion of the crude oil from being effectively displaced, resulting in residual oil remaining in the reservoir, especially in complex and narrow pore structures. As a result, the permeability of the reservoir decreases significantly, thereby negatively impacting the overall oilfield development performance.

More importantly, the formation of flocculated precipitates not only affects the physical properties of the reservoir but may also alter the rock wettability, leading to changes in the wettability state of the reservoir. These changes in wettability could cause uneven oil–water distribution, further affecting oil and gas production efficiency. Additionally, the reduced fluidity, pore blockage, and residual oil caused by flocculation reactions during polymer flooding increase the energy consumption required for oil displacement, thereby raising the displacement cost and reducing the economic efficiency of the compound flooding process.

This series of negative effects not only directly impacts recovery efficiency but may also jeopardize the sustainability of the entire oil displacement process. Therefore, understanding the flocculation reactions between polymers and kaolinite, and their effects during the oil displacement process, is crucial for optimizing compound flooding technologies, improving recovery rates, reducing energy consumption, and enhancing economic benefits. Through further experiments and modeling studies, it may be possible to explore ways to control or reduce the occurrence of these flocculation reactions, thereby improving the effectiveness of polymer flooding and achieving more efficient and economical oilfield development.

To optimize the oil recovery process and reduce the negative effects of kaolinite flocculation, effective measures need to be implemented during the enhanced oil recovery process. First, selecting the appropriate drive agents and using anti-flocculants can effectively prevent flocculation, ensuring the flowability of the displacing fluid. Additionally, post-treatment methods such as acid fracturing can remove deposits in the pore spaces, restoring the permeability of the reservoir. Furthermore, improving the formulation of injection fluids, optimizing the concentration and type of polymers, and combining reasonable injection plans can help minimize the damage caused by flocculation to the reservoir. By implementing these measures, not only can the recovery rate be improved, but the oil recovery cost can also be reduced, enhancing the overall development efficiency of the oil field.

## 5. The Distribution of Residual Oil After Binary Composite Flooding

Based on the distribution of residual oil and water, the first type is free-phase residual oil, which includes the following subtypes: intragranular, faint mist-like, intergranular adsorption, and cluster-like residual oil. The second type is semi-bound residual oil, which refers to residual oil that is either in the outer layer or relatively far from the mineral surface. This includes subtypes such as precipitated pores, corner spaces, and throats. The third type is adsorbed residual oil, which is bound to the mineral surface, and is further classified into particle-adsorbed, narrow gap, and thin film on pore surfaces residual oil [29,30,31,32]. 

After composite flooding, the residual oil mainly consisted of six types, corresponding to the different states of oil retention in the porous medium (see Figure 12).

After binary composite flooding, the microscopic distribution characteristics of residual oil exhibit an overall scattered distribution with local enrichment zones. As the injection volume increases, the content of free-phase residual oil decreases, while the content of bound-phase residual oil increases. This portion of residual oil is difficult to mobilize [33,34].

In this experiment, as a micro-scale displacement visualization model was used to simulate the real reservoir. The residual oil types mainly consisted of four types: cluster-type, throat-type, pore surface film-type, and corner-type residual oil (see Figure 13).

## 6. Conclusions

When preparing the model, the adhesive should not be applied too thickly, and care must be taken during development to avoid exposing the glass bottom surface. The UV exposure time should also be carefully controlled, as too long an exposure can affect the hardness of the adhesive layer. After exposure, the model must be immediately placed on a hot plate for curing; failure to do so within a reasonable time may cause image fading. To ensure even heating, a thick metal plate should be placed on the hot plate, as excessively high temperatures can damage the adhesive’s properties. Additionally, the immersion time in the development solution should not be prolonged, as excessive immersion can cause the adhesive to detach. It is important to keep these factors in mind to maintain the integrity of the model during the process.

The particle migration phenomenon occurs during polymer flooding, where the polymer, acting as a flocculant, draws suspended particles closer through electrostatic neutralization, adsorption bridging, compression of the double electric layer, and network capture. This interaction leads to the migration of the suspended particles, which may be adsorbed onto the surface of the flocculant molecules or encapsulated in a “core–shell” structure, eventually forming precipitates that block pore throats or remain trapped in the pores. The flocculation effect of the polymer causes the migration of clay particles in the formation, which may lead to blockages in polymer flooding or polymer-led composite flooding. Furthermore, when mixed with crude oil, a high-viscosity oil mixture is formed, causing crude oil that could previously be easily recovered to become trapped due to increased viscosity. To improve the recovery rate of binary composite flooding, effective measures should be taken, including selecting appropriate flooding agents, using anti-flocculants, employing acid fracturing and other post-treatment methods, optimizing injection fluid formulations, and refining injection strategies.

## Figures and Tables

**Figure 1 polymers-16-03494-f001:**
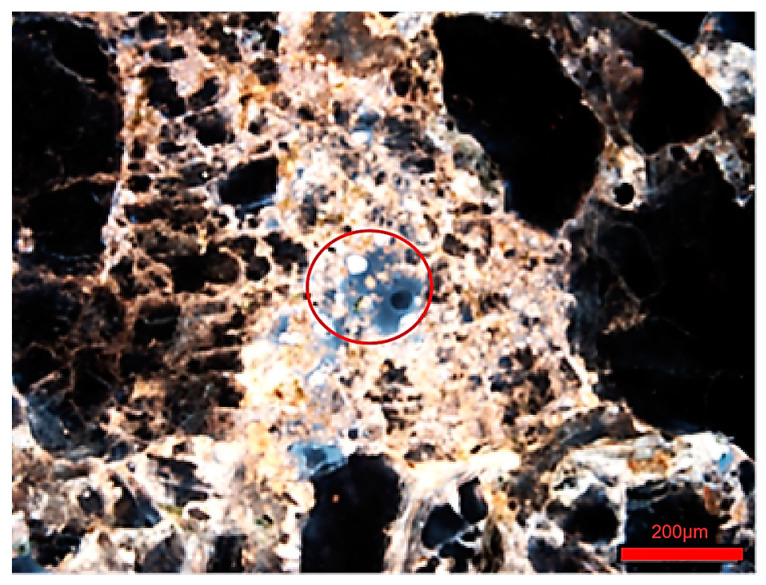
Inter-particle adsorption of residual oil diagram.

**Figure 2 polymers-16-03494-f002:**
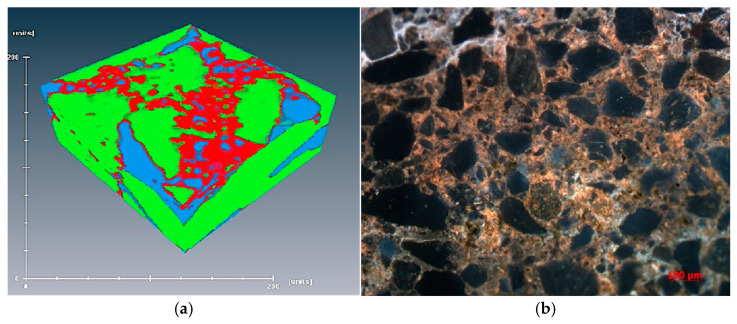
Inter-particle adsorption of residual oil in (**a**) CT 3D image and (**b**) fluorescence photograph.

**Figure 3 polymers-16-03494-f003:**
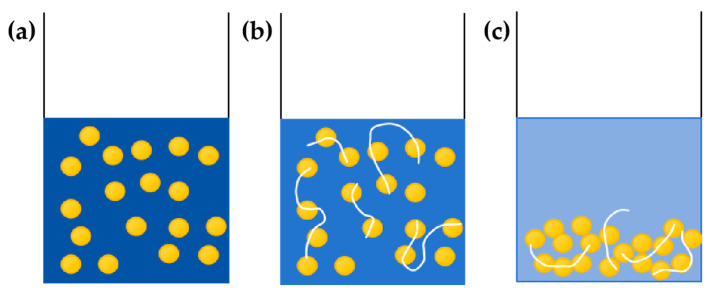
Flocculation diagram. (**a**) Kaolinite water suspension. (**b**) Formation of flocs. (**c**) Precipitation.

**Figure 4 polymers-16-03494-f004:**
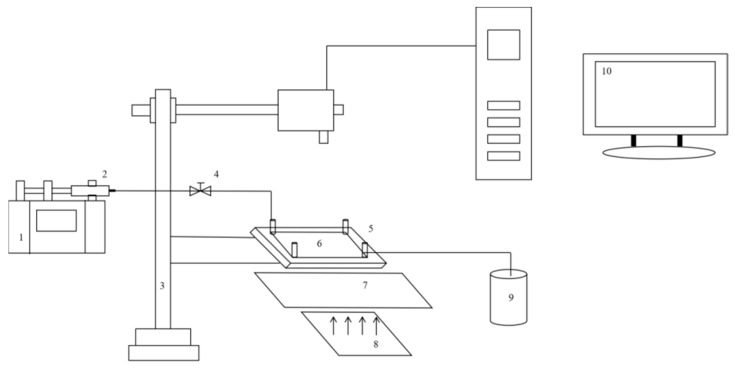
Experimental apparatus diagram. 1 Micro-injection pump. 2 Injection syringe. 3 Microscope. 4 Valve. 5 Micro-visualization model holder. 6 Micro-visualization model. 7 Heating plate. 8 Parallel light source. 9 Liquid recovery collection container. 10 Image acquisition computer.

**Figure 5 polymers-16-03494-f005:**
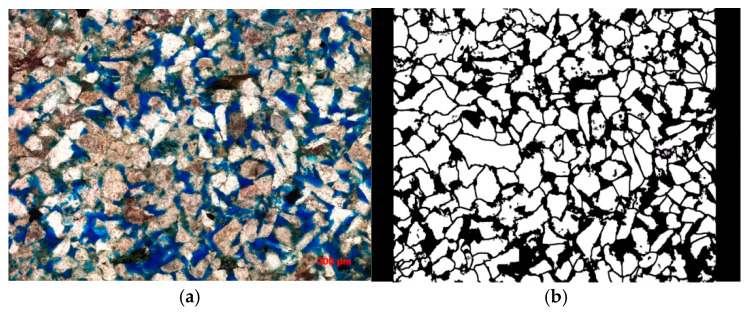
Cast thin sheet. (**a**) Laser confocal scanning injection thin sheet; (**b**) vectorized image after vectorization by software(Imagine v1.7.1).

**Figure 6 polymers-16-03494-f006:**
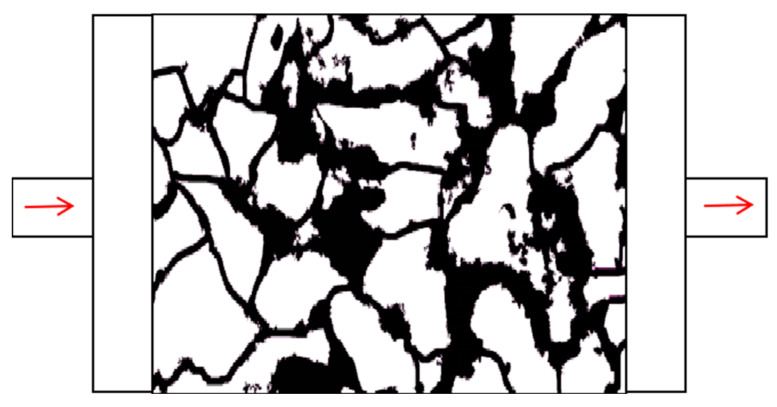
Microscopic displacement etching model.

**Figure 7 polymers-16-03494-f007:**
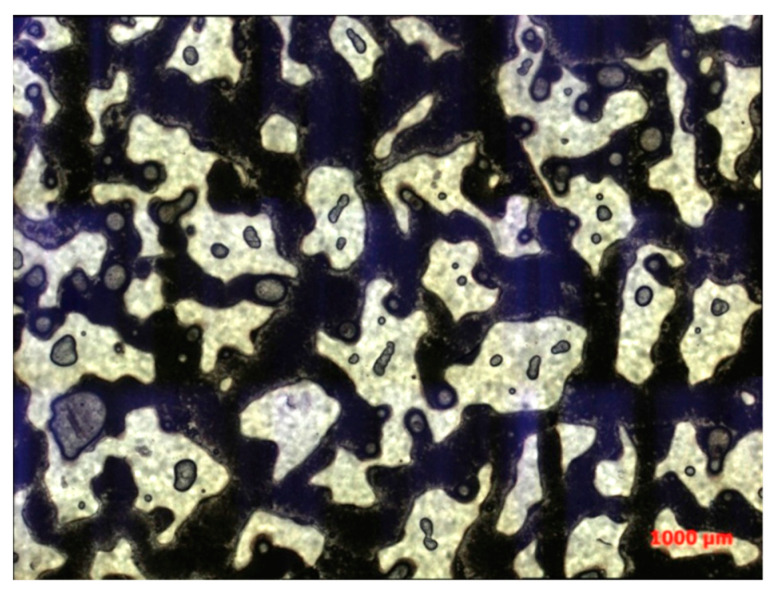
Kaolinite aqueous suspension (30% concentration), transmitted light photograph, temperature 46 °C.

**Figure 8 polymers-16-03494-f008:**
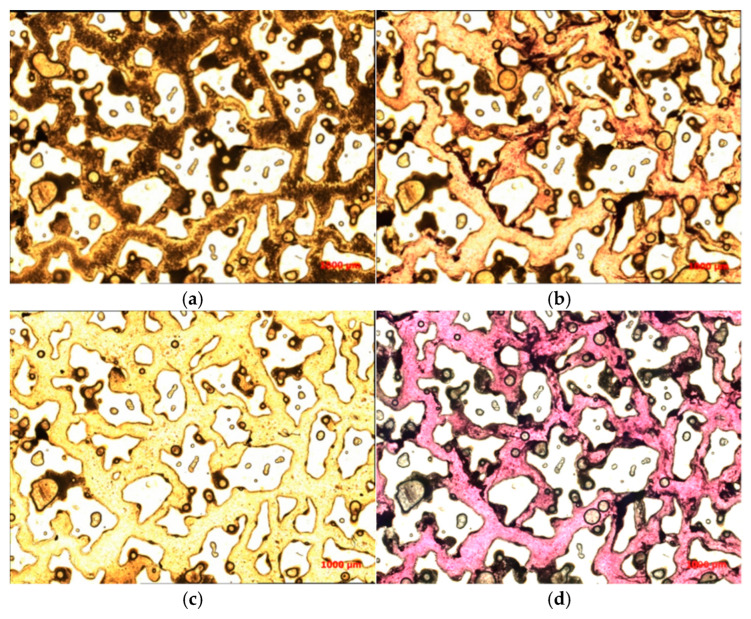
The oil displacement effect images from Experiment 1 are as follows. (**a**) Waterflooding (10 PV) transmitted light photograph. (**b**) Binary composite flooding (10 PV) transmitted light photograph. (**c**) Waterflooding (20 PV) transmitted light photograph. (**d**) Binary composite flooding (20 PV) transmitted light photograph.

**Figure 9 polymers-16-03494-f009:**
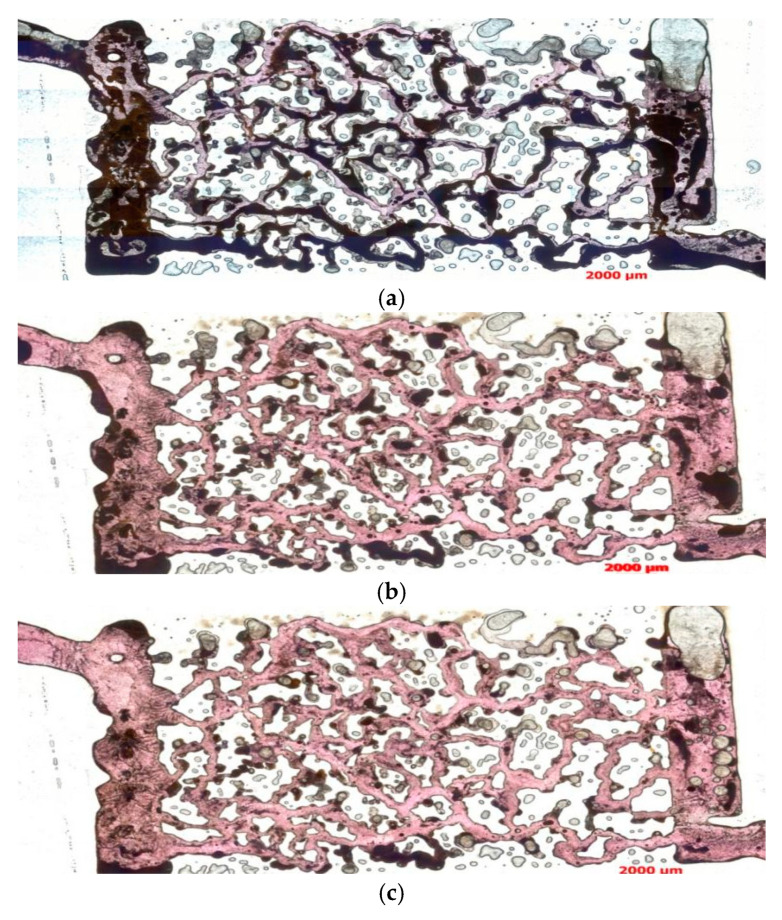
The oil displacement effect images from Experiment 2 are as follows. (**a**) Binary composite flooding (10 PV) displacement effect. (**b**) Binary composite flooding (20 PV) displacement effect. (**c**) Binary composite flooding (80 PV) displacement effect.

**Figure 10 polymers-16-03494-f010:**
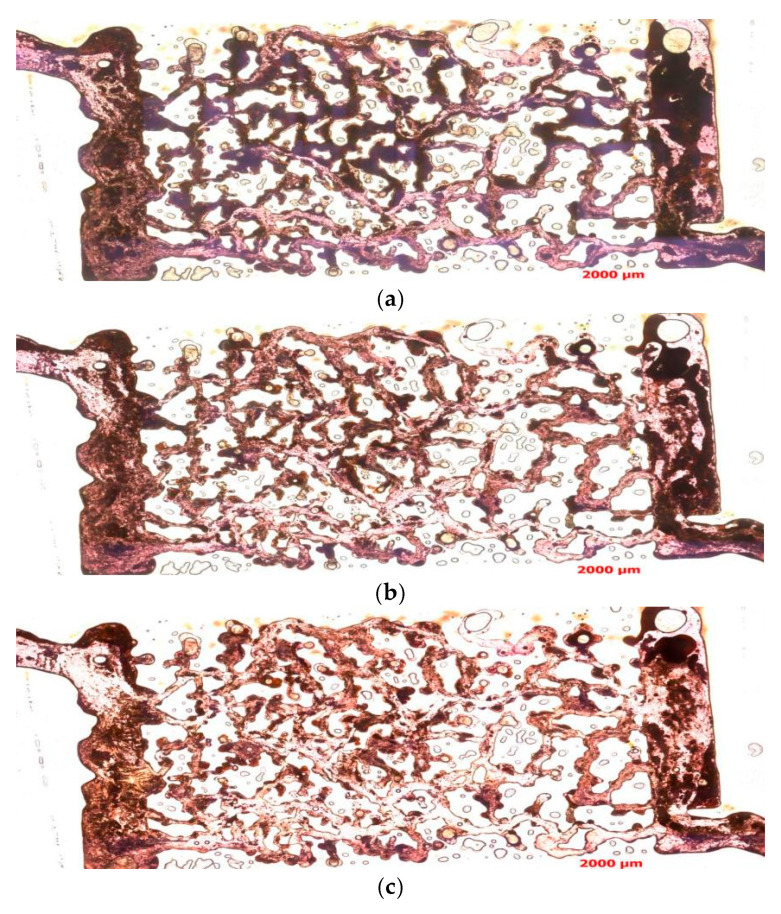
The oil displacement effect images from Experiment 3 are as follows. (**a**) Binary composite flooding (10 PV) displacement effect. (**b**) Binary composite flooding (20 PV) displacement effect. (**c**) Binary composite flooding (80 PV) displacement effect.

**Figure 11 polymers-16-03494-f011:**
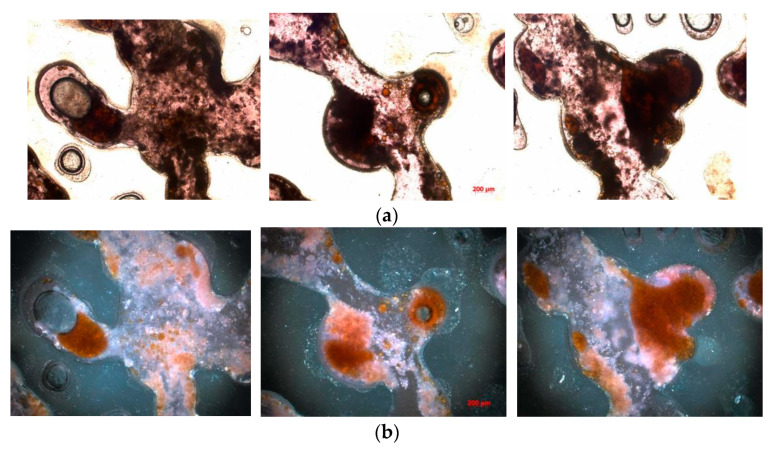
The distribution of the mixed solution is shown as follows: (**a**) transmitted light photograph and (**b**) orthogonal light photograph.

**Figure 12 polymers-16-03494-f012:**
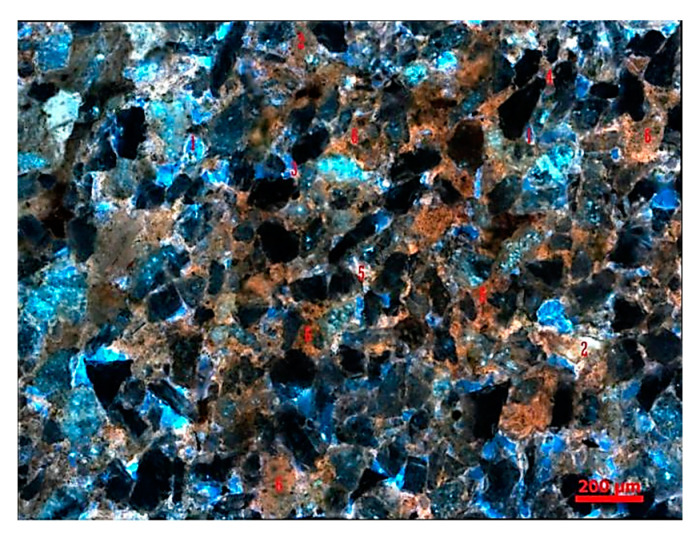
The schematic of residual oil distribution types is as follows. 1 Pore surface film-type. 2 Grain adsorption-type. 3 Corner-type. 4 Throat-type. 5 Cluster-type. 6 Intergranular adsorption-type.

**Figure 13 polymers-16-03494-f013:**
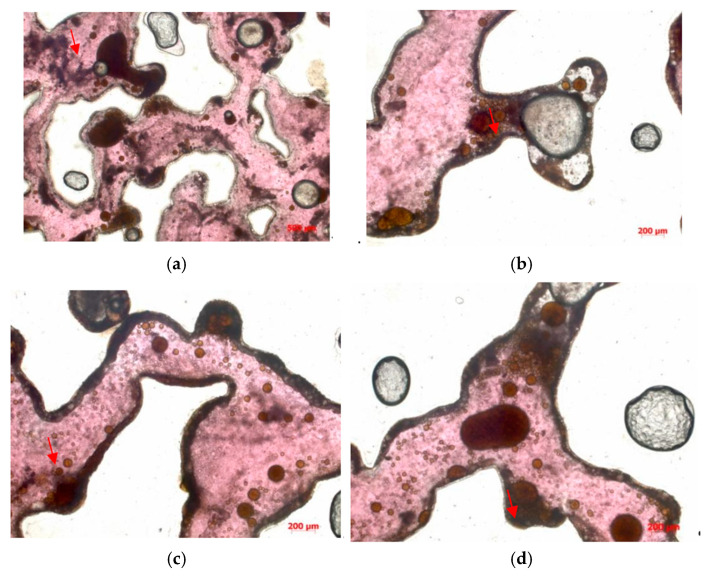
The residual oil distribution types are as follows. (**a**) Cluster-type. (**b**) Throat-type. (**c**) Pore surface film-type. (**d**) Corner-type. (The arrow points to the location where the residual oil is located).

## Data Availability

The original contributions presented in this study are included in the article. Further inquiries can be directed to the corresponding author.

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
