# Peer review of "Investigating the Impact of Polymers on Clay Flocculation and Residual Oil Behaviour Using a 2.5D Model"

_polymers, 2024, doi:10.3390/polym16243494_

Round 1
Reviewer 1 Report
Comments and Suggestions for Authors
This study aims to employ a microvisual displacement model and microscopy to analyse and observe these phenomena. The research findings are impactful to the oilfield development. Here are some specific comments:
-Introduction: Now, the introduction section is written briefly, but it can be elaborated more in detail.
-You should add the sub-title " Methodology" to clearly show which part is telling the method of your research.
- The graph and figure have been shown in a clear way to support the results.
-Conclusion: It should be written in a paragraph or point form to show your study's significant findings clearly. It is not written in a professional format, although it captures the results and explanation of your findings. You may also consider adding one sentence at the end to explain your study's limitations and potential application.
Author Response
please see the details in the attachment

Reviewer 2 Report
Comments and Suggestions for Authors
The manuscript lacks sufficient clarity in explaining the 2.5D model and could benefit from deeper analysis and comparison with existing studies and revising the manuscript in the light of following comments:
1. Consider rephrasing the title for more clarity, such as "Investigating the Impact of Polymers on Clay Flocculation and Residual Oil Behaviour Using a 2.5D Model.
2. The introduction should adequately present the context and relevance of studying polymer effects on clay flocculation and residual oil. However, the link between the polymer properties and their role in these processes should be better established
3. Provide a more comprehensive literature review to strengthen the background, especially focusing on previous studies and why the 2.5D model is an appropriate choice for this study.
4. The use of the 2.5D model should be clearly defined. What exactly does this model represent in the context of clay flocculation and residual oil? The paper needs to explain this model's advantages and limitations
5. The procedure for polymer preparation, clay suspension, and residual oil measurement could be elaborated for reproducibility
6. Provide more details on the experimental setup, including the type of polymers used, their concentrations, and specific techniques for observing clay flocculation and residual oil.
7. The data is presented in graphs and tables, but the interpretation of these results could be more in-depth. There should be more emphasis on the relationship between polymer concentration and clay behaviour, and how this influences residual oil.
8. The results should be compared with previous research to highlight any new insights or contradictions
9. Reorganize the section to first present the key findings in a concise manner, followed by a thorough analysis of how polymers affect clay flocculation and residual oil
10. The implications of the research for future studies or industrial applications, especially in oil recovery, should be clearer
11. Include captions that provide brief descriptions of the significance of each figure or table, ensuring that the reader can understand them without needing to refer to the main text constantly.
12. Update the references, especially in relation to polymer-clay interactions and the application of the 2.5D model in similar studies
Comments on the Quality of English Language
English Language needs thorough corrections
Author Response

(The authors gave the same response as above.)

Round 2
Reviewer 2 Report
Comments and Suggestions for Authors
The authors revised the manuscript successfully and may be accepted